# The Extra Cost Due to Non-Adherence to Inhaled Treatments in Adolescents with Mild-to-Moderate Persistent Asthma

**DOI:** 10.3390/children10040615

**Published:** 2023-03-24

**Authors:** Roberto Walter Dal Negro, Paola Turco

**Affiliations:** National Centre for Respiratory Pharmacoeconomics and Pharmacoepidemiology (CESFAR), 37124 Verona, Italy

**Keywords:** adolescents, mild-to-moderate asthma, inhaled treatments, non-adherence, cost

## Abstract

Bronchial asthma has a high socio-economic impact in Western countries. Low adherence to prescribed inhalation treatments contributes to poor asthma control and the higher utilization of healthcare resources. Although adolescents usually do not comply with long-term inhaled treatments prescribed on a regular basis, the related economic consequences still are poorly investigated in Italy. Aim: A 12-month estimation of the economic impact of non-adherence to inhalation treatments in adolescents with mild-to-moderate atopic asthma. Methods: Non-smoking adolescents aged 12–19 years, without any significant comorbidity, prescribed with inhaled cortico-steroids (ICS) or ICS/long-acting beta(2)-adrenergics (LABA) via dry powder inhalers (DPIs) on a regular basis were automatically selected from the institutional database. Spirometric lung function, clinical outcomes, and pharmacological information were collected. The adolescents’ adherence to their prescribed regimen was calculated monthly. Adolescents were divided in two sub-groups based on their adherence to prescriptions: ≤70% (not adherent) or >70% (adherent), and statistically compared (Wilcoxon test, assuming *p* < 0.05). Results: Overall, 155 adolescents fulfilled the inclusion criteria (males, 49.0%; mean age, 15.6 years ± 2.9 SD; mean BMI, 19.1 ± 1.3 SD). Mean values of lung function were: FEV1 = 84.9% pred. ± 14.8 SD, FEV1/FVC = 87.9 ± 12.5 SD; MMEF = 74.8% pred. ± 15.1 SD and V25 = 68.4% pred. ± 14.9 SD. ICS had been prescribed in 57.4% of subjects and ICS/LABA in 42.6%. Mean adherence to original prescriptions was 46.6% ± 9.2 SD in non-adherent and 80.3% ± 6.6 SD in adherent adolescents, respectively (*p* < 0.001). The mean rates of hospitalizations, exacerbations, and GP visits; the average duration of absenteeism; the frequency of systemic steroids and antibiotics courses needed over the study period were significantly and substantially lower in adolescents adherent to prescriptions (all *p* < 0.001). The mean total annual extra cost calculated in the two sub-groups was EUR 705.8 ± 420.9 SD in non-adherent adolescents and EUR 192.1 ± 68.1 SD in adherent adolescents, respectively (*p* < 0.001), which was 3.7 times higher than in non-adherent adolescents. Conclusions: In adolescents, the clinical control of mild-to-moderate atopic asthma is directly and strictly related to the degree of adherence to prescribed inhalation therapies. All clinical and economic outcomes prove dramatically poor when adherence is low, and treatable asthma can be frequently mistaken for refractory asthma in these cases. Adolescents’ non-adherence impacts the burden of the disease quite substantially. Much more effective strategies centered specifically on adolescents’ asthma are needed.

## 1. Background

Bronchial asthma is a chronic disease of the airways highly impacting on patients, their families, and society as a whole [1]. Airway inflammation is the main pathogenetic determinant: this condition is able to induce the occurrence of variable airway obstruction (mostly reversible) with several respiratory symptoms (namely wheeze, shortness of breath, chest tightness, and/or cough). The frequency of symptoms, together with lung function (usually the forced expiratory volume in 1 s—FEV1), the variability of peak expiratory flow (PEF), and the frequency of exacerbations contribute to defining the severity of asthma (such as mild intermittent, mild persistent, moderate persistent, and severe persistent [1].

Bronchial asthma also represents a high socio-economic burden in Italy [2]. Further to direct costs (mainly related to hospitalizations, exacerbations, and drugs), the loss of productivity in adults and school absenteeism in children and adolescents (i.e., indirect costs) contribute significantly to the annual impact of asthma [3].

Patients’ sub-optimal adherence to prescribed inhalation treatments is presumed to affect the impact of the disease even when asthma is of mild-to-moderate severity [4,5,6]. It has generally been accepted since long ago that clinical outcomes, medium- and long-term asthma control, and quality of life are poor when patients comply ≤70% with their prescriptions [7]. This aspect represents a critical issue, particularly in children and adolescents, due to their much more frequent suboptimal adherence to long-term inhaled treatments [8,9], sometimes even independent of the complexity of the therapeutic regimen adopted [10].

The aim of the study was the 12-month estimation of the economic impact of insufficient adherence to inhalation treatments in adolescents with mild-to-moderate atopic asthma.

## 2. Methods

The study was a retrospective analysis of asthmatic adolescents referred to the Lung Unit of the Specialist Medical Centre (CEMS), Verona, Italy, over the period June 2019–May 2020. Patient selection was performed anonymously from the institutional database (UNI EN ISO 9001 validated and presently containing data from around 100,000 respiratory patients) by Boolean algebraic formula [11]. All patients’ records provide anagraphics; body mass index (BMI); anamnestic and clinical data; lung function parameters; analytic therapeutic prescriptions; duration of treatments; general acceptability of treatments; and health economic data; all were continuously updated.

Basic selection criteria were: adolescents of both genders ranging 12–19 years of age with mild-to-moderate asthma, non-smoker, with a normal cognitive function and without any relevant comorbidity, who had been prescribed an inhaled corticosteroid (ICS) alone or in combination with a long-acting β-adrenergic (ICS/LABA) via dry powder inhalers (DPIs). Exclusion criteria were: the presence of physical and neurological conditions that made the inhalation difficult or improper; the non-availability of a complete set of data required by the study plan; the patient’s (and/or his/her parent’s) refusal to participate in the study.

As all DPIs contain doses of respiratory drug(s) to inhale enough for thirty/sixty days of treatment and are all provided with a dose counter, the adolescents’ adherence to the inhalation regimen prescribed was calculated at the end of every month via a telephone call during which each patient (or one of his/her parents) had to communicate the number of remaining doses visible in the device to the interviewer. The compliance was then calculated as % skipped inhalation doses/every month, which mirrored the average skipping days of treatment over the study period [10].

Lung function parameters collected at recruitment were: Forced Expiratory Volume in 1 s (FEV1); FEV1/Forced Vital Capacity (FEV1/FVC); Maximum Mid-Expiratory Flow (MMEF); and Maximum Expiratory Flow at 25% of lung filling (V25). Spirometric measures were obtained by means of the MiniBox^TM^ (PulmOne Advanced Medical Devices, Ltd., Ra’anana, Israel) and reported as % predicted values.

Clinical outcomes collected over the study period were the hospitalization rate; the duration of hospital stay (in days); the absenteeism duration (in days); the exacerbation rate; the n. of GP (or specialist) visits; the n. of systemic steroids and of antibiotics courses prescribed over the study period. Their corresponding mean costs were calculated and reported in Euros [12]. As the study did not aim to compare the efficacy of different respiratory drugs, the cost of prescribed regular daily treatments was equalized and not included in the present study in order to only assess the extra impact of different degrees of adherence.

Patients recruited were divided in two sub-groups according to the degree of their adherence to the therapeutic regimen originally prescribed: ≤70% (non-adherent) or >70% (adherent) of prescribed daily doses. All data collected from adherent and non-adherent adolescents were reported as means ± SD and compared by Wilcoxon test; *p* < 0.05 was accepted as the statistical significancy limit.

Ethics: The study was approved by the Ethical Committee during the session officially held on October 10 (code: 02/RG02/2017). Even though the selection of subjects was conducted anonymously and by automatic procedures from the database, informed consent from the adolescents’ parents was requested for the scientific use of data collected.

## 3. Results

The whole sample of subjects fulfilling inclusion criteria consisted of 155 adolescents with mild-to-moderate atopic asthma (76 males, 49.0%), mean age = 15.6 ± 2.9, and mean BMI = 19.1 ± 1.3. Of those, 69 (44.6%) were only sensitized to seasonal allergens, 34 (21.9%) only to perennial allergens, and 52 (33.5%) to both. Mean spirometric data (reported as % predicted) were: FEV1 = 84.9% ± 14.8 SD; FEV1/FVC = 87.9% ± 12.5 SD; MMEF = 74.8% ± 15.1 SD; and V25 = 68.4% ± 14.9 SD. Patients’ original prescriptions were ICS only in 89 patients (57.4%) and ICS/LABA in 66 patients (42.6%).

Data for the whole sample and data from both adherent and non-adherent adolescents are reported in Table 1 together with the corresponding statistical significance of comparisons. Adherent and non-adherent asthmatic adolescents proved well-matched in terms of anagraphics, basal lung function, and sensitization to allergens. The distribution of ICS and ICS/LABA prescriptions were also well matched.

Mean adherence to treatments calculated in the whole sample was 63.4% ± 8.1 SD, without any significant difference for age and gender. In general, mean adherence to a once-daily regimen via DPIs proved slightly higher, even if not significantly different from that assessed with the twice-daily strategy (64.4% ± 9.3 SD vs. 62.8% ± 8.7 SD; *p* = ns). Mean adherence calculated in adolescents treated with only ICS was 63.0% ± 9.1 SD while that one in those treated with ICS/LABA was 62.4% ± 8.3 SD, which was absolutely comparable (*p* = ns).

Adherent and non-adherent adolescents proved only significantly different in terms of degree of long-term adherence to their original respiratory prescriptions, such as 46.6% ± 9.2 SD in non-adherent adolescents vs. 80.3% ± 6.6 SD in adherent adolescents, respectively (*p* < 0.001) (Table 1).

Mean values ± SD of all outcomes collected in the two sub-groups are reported in Table 2 together with the corresponding statistical comparisons. All outcomes proved dramatically lower in adolescents showing poor adherence to their prescriptions (all *p* < 0.001) (Table 2).

Moreover, the mean cost of each clinical outcome considered proved significantly higher in non-adherent adolescents, both in terms of direct and indirect costs (Table 3). No significant differences were found for age and gender.

Finally, the mean total annual extra cost calculated in the two sub-groups was EUR 705.8 ± 420.9 SD in non-adherent adolescents and EUR 192.1 ± 68.1 SD in adherent adolescents (*p* < 0.001). The difference in the economic impact between the two sub-groups proved substantial and highly significant (Figure 1).

## 4. Discussion

The aim of the present study was to focus primarily the clinical and the related economic consequences of sub-optimal adherence to inhalation treatments prescribed in adolescents with mild-to-moderate bronchial asthma. This topic has not been frequently investigated, even though poor adherence is presumed to cause a quite poor effectiveness of whatever inhalation strategy in real-life.

Asthma of milder severity is usually under-recognized and under-estimated even though specific studies underlined that early and careful pharmacological interventions are likely to lead to more effective control and/or can prevent the progressive worsening of asthma [13,14].

The adherence to inhalation treatments still is regarded as a priority in the clinical governance of asthma and its periodical check has been strongly recommended since long ago [15]. Regardless of the specific cause(s) of their non-adherence (namely, cultural, educational, behavioral, or psychological in origin) [16,17,18,19,20,21,22], younger patients (in particular adolescents) when compared to adults usually do not comply sufficiently with the regular strategy and the proper inhalation procedures required for effective asthma management [8,9,23,24]: actually, poor physiological and clinical long-term outcomes are associated with these cases [10].

Several studies investigated the cost-effectiveness of therapeutic intervention against asthma in the last decades, but only a negligible proportion of these studies specifically assessed the relative role of sub-optimal adherence to inhalation treatments [25]. However, although almost exclusively assessed in adults with asthma of higher severity, a strict negative correlation between non-adherence, general healthcare utilization, and associated costs has been described [26,27].

This peculiar aspect was much less investigated in adolescents, particularly in those with milder degrees of asthma. A few years ago, a study carried out on a population of more than 2000 asthma patients aged ≥15 years reported that only 51% of patients were adherent to treatment and that the annual cost of non-adherent patients was twice that of adherent patients (such as: EUR 1431 vs. 722 [28]. Unfortunately, patients included in that study were moderate asthmatics in the great majority of cases and the study was not only dedicated to adolescents.

Data from the present study stem from a twelve-month survey centered exclusively on adolescents with mild-to-moderate persistent asthma. As in previous experiences [15,28,29,30,31], the mean global adherence of young asthmatics to prescribed respiratory treatments was confirmed generally low in the present study (around 60%). However, all clinical outcomes resulted dramatically in favor of adolescents who showed a ≥70% adherence to their prescribed inhalation therapy. Indeed, exacerbation and hospitalization rates, together with the need for extra prescriptions of systemic steroids, antibiotics, and rescue short-term bronchodilators were minimized over the study period when the adherence was acceptable. All components of direct and indirect costs (namely, the impact of school absenteeism) were also much lower in these cases. In other words, the extra cost only induced by non-adherence proved 3.7 times higher when compared to that calculated when adherence is acceptable.

The present study focused and emphasized for the first time on our knowledge the relative economic extra impact of non-adherence to inhaled treatments in adolescents with milder forms of asthma. It should also be considered that a high prevalence of poor adherence can contribute dangerously to the misidentification of treatable milder forms of asthma as refractory asthma, without negligible social and economic related consequences.

Some limitations of the present study are: (a) the use of a monocentric model for the study; (b) the limited sample size; (c) respiratory drugs for inhalation were only prescribed via DPIs in order to facilitate the monthly calculation of adolescents’ adherence at home; (d) once-daily and twice-daily inhalation regimens were pooled in the present study due to the age-dependent limitations (in the range 12–18 years of age) of prescription for different ICS and LABA molecules in our country; (e) the effectiveness of inhalation maneuvers was impossible to be checked at home. Possible points of strength are: (a) the sample of adolescents was obtained automatically from the central database by means of anonymous pre-defined Boolean equations strictly corresponding to the inclusion and exclusion criteria; (b) the cost of original prescriptions was equalized in order to only assess the extra impact of non-adherence over the study period; (c) the survey focused specifically the cost of non-adherence in adolescents with milder forms of asthma, which was previously missing.

## 5. Conclusions

Adolescents suffering from milder forms of asthma were once again characterized by their poor adherence to regular inhalation treatments over long periods. The trends in asthma control were confirmed to be strictly and directly related to the degree of adherence to inhalation therapies. Clinical and economic outcomes are dramatically poor when adherence is low, thus reflecting much lower long-term control of asthma. Poor adherence can lead to the erroneous identification of easily treatable mild-to-moderate asthma as refractory asthma, thus requiring much more expensive therapeutic strategies. Adolescents’ non-adherence impacts the economic burden of the disease substantially. Much more effective strategies, specifically centered on mild-to-moderate adolescents’ asthma, are needed.

## Figures and Tables

**Figure 1 children-10-00615-f001:**
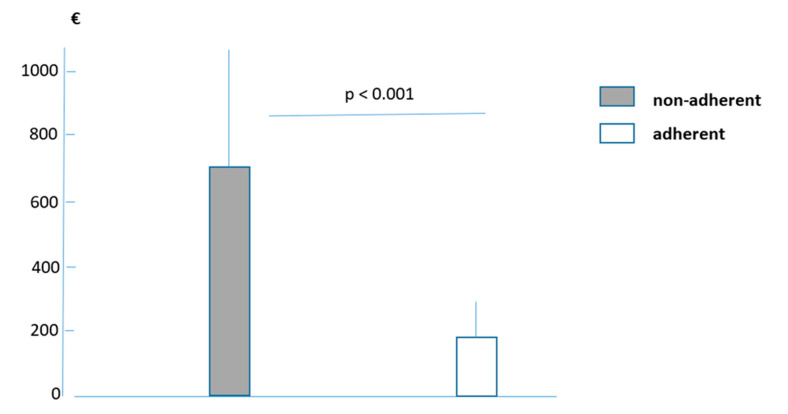
Mean cost (means ± SD) in Euros (€) over twelve months in non-adherent and adherent adolescents, and the corresponding statistical comparison (Wilcoxon test). (Cost of prescribed regular daily treatment not included.)

**Table 1 children-10-00615-t001:** General characteristics of the sample.

	Total Sample	Non-Adherent	Adherent	*p*
Subjects (*n*)	155	72	83	
males (*n*; %)	76; 49.0	37; 51.3	41; 49.4	
age in yrs (mean ± SD)	15.6 ± 2.9	15.2 ± 3.1	15.8 ± 3.6	ns
BMI (means ± SD)	19.1 ± 1.3	18.8 ± 1.6	19.4 ± 1.9	ns
Allergens				
seasonal only (*n*; %)	69; 44.6	29; 40.3	35; 42.2	ns
perennial only (*n*; %)	34; 21.9	17; 23.6	19; 22.9	ns
mixed (*n*; %)	52; 33.5	26; 36.1	29; 34.9	ns
Lung function				
FEV1 % pred. (means ± SD)	84.9 ± 14.8	85.4 ± 14.1	84.2 ± 15.5	ns
FEV1/FVC % pred. (mean ± SD)	87.9 ± 12.5	88.7 ±11.9	85.8 ± 13.4	ns
MMEF % pred. (means ± SD)	74.8 ± 15.1	75.9 ± 19.4	77.4 ± 17.3	ns
V25 % pred. (means ± SD)	68.4 ± 14.9	66.1 ± 19.5	67.8 ± 18.2	ns
Basal prescriptions				
ICS DPIs (*n*; %)	89; 57.4	41; 56.9	48; 57.8	ns
ICS/LABA DPIs (*n*; %)	66; 42.6	31; 43.1	35; 42.2	ns
Adherence to treatment (%)	63.4 ± 8.1	46.6 ± 9.2	80.3 ± 6.6	0.001

FEV1: Forced Expiratory Volume in 1 s; FEV1/FVC: Forced Expiratory Volume in 1 s/Forced Vital Capacity; MMEF: Maximum Mid Expiratory Flow; V25: Maximum Expiratory Flow at 25% of lung filling. ICS: inhaled corticosteroid; LABA: long-acting beta2-adrenergic; DPI: dry powder inhaler. (Wilcoxon test).

**Table 2 children-10-00615-t002:** Outcomes in non-adherent and adherent adolescents over 12 months, and corresponding statistical comparisons (means ± SD).

	Non-Adherent	Adherent	
Outcomes	12 Months	12 Months	*p*
Hospitalizations (*n*)	1.2 ± 0.9	0.1 ± 0.6	0.001
Duration (days)	1.6 ± 1.7	0.3 ± 0.7	0.001
Absenteeism (days)	6.5 ± 3.3	0.2 ± 1.6	0.001
Exacerbations (*n*)	1.4 ± 1.8	0.3 ± 1.3	0.001
Visits (*n*)	2.6 ± 2.3	0.8 ± 1.7	0.001
Courses of syst. steroids (*n*)	3.1 ± 1.7	0.2 ± 0.3	0.001
Courses of antibiotics (*n*)	2.4 ± 2.2	0.2 ± 0.6	0.001

(Wilcoxon test).

**Table 3 children-10-00615-t003:** Components of mean cost (means ± SD) over twelve months in non-adherent and adherent adolescents, and corresponding statistical comparisons. (Cost of prescribed regular daily treatment not included.)

	Non-Adherent	Adherent	
Cost (EUR)			*p*
Hospitalizations	328.2 ± 217.4	86.6 ± 111.3	0.001
Visits	136.6 ± 43.6	26.8 ± 14.2	0.001
Indirect costs	127.4 ± 86.6	46.5 ± 20.3	0.02
Systemic steroids	31.1 ± 22.3	10.1 ± 5.3	0.001
Antibiotics	35.6 ± 26.7	9.5 ± 6.6	0.001
Rescue drugs	46.9 ± 24.3	12.6 ± 8.4	0.001

(Wilcoxon test).

## Data Availability

Data are available in the CESFAR data archives.

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
