# Peer review of "The Extra Cost Due to Non-Adherence to Inhaled Treatments in Adolescents with Mild-to-Moderate Persistent Asthma"

_children, 2023, doi:10.3390/children10040615_

Round 1

Reviewer 1 Report

Manuscript: The Extra-Cost due to the Non-Adherence to Inhaled Treatments in Adolescents with Mild-to-Moderate Persistent Asthma

This is an interesting and well-written manuscript. The study is original and useful in taking into account the costs of treatment and the consequences of the lack of adherence to the treatment of asthma in adolescents.

The study has several limitations:

Major:

Methods: why the Tiffanou index (FEV1/VC), which is the basic parameter determined in asthma, is not given, please complete this in the methodology and results.

Discussion: – please write 2-3 sentences at the beginning of the discussion about what the work is, what it concerns and what are the most important and interesting discoveries of the study.

Table 1- Please separate the results of the subsections of “Allergens” and “basal prescriptions”  more clearly, e.g. 69 (44.6%)  

Minor

abstract:  Please rewrite more clearly on the section methods and results

The work requires language and editorial  correction (for example line 186 tudy=study)

Author Response

Thankyou very much indeed to Reviewer #1 for defining the manuscript "interesting, well-written, original and useful in taking into account the costs of lack of adherence to treatments.

Major criticisms:

1) "why the Tiffenau ....": according to your sugegstion, the corresponding values +/-sd have beed added in the Abstract, Methods, and Results sections, and also in table1.

2) A couple of sentences have been added at the beginning of the Discussion

3) according to your suggestion, data of Allergens and Basal prescriptions have beed separated in Table 1

Minor criticisms: 

1) methods and results in in Abstract section have been re-writed more clealy

2) line 186 : corrected

All change are visible in RED

Reviewer 2 Report

Comments to the authors

Summary: This retrospective study compared the economic cost of asthma in adolescent patients with mild-to-moderate asthma adhering and non-adhering to pharmacotherapy. Data analysis revealed that the economic cost of asthma was over 3-fold higher in non-adherents versus adherents. The authors conclude that more strategies targeting adolescents are warranted to improve asthma control in this age group.

Overall comment

Indeed, suffering from asthma causes a physical and mental toll on children of all ages along with a financial burden for the family and society. Therefore identifying approaches to improve asthma control in children is of great importance on an individual and public health level.

This article was well-written and contributes new evidence to the existing database. The authors are to be commended. Nevertheless, there are minor issues that require revision.

We look forward to future studies from this research group.

Comments:

Abstract

Line 14, ICS etc to be written in full when it appears the first time in the text

Line 16 Indicate how lung function was assessed e.g spirometry

Line 20 FEV1, 1 should be written as a subscript throughout the manuscript FEV1

Line 21 sd written as S.D or s.d

Line 29, typographical error. ‘ In adolescents’ not in adolescents, the clinical control of……

Main text

Line 56, typographical error ‘ 12-month estimation’  not extimation

Line 87 ‘Clinical collection……” text size is not consistent

Line 106 Spelling error ‘fullfilling’

Result: Below all tables indicate what statistical test was used to estimate P-values

Line 134, Table 2 legend means ± S.D not ds

Table 2 If I am not mistaken, the same data are presented 3 times.

If the outcome was assessed at 3 time points, this is not clear from the methods section and should be mentioned in Table 2. Please clarify.

Line 145, delete figure 1 since the total economic cost is mentioned in the text and is obvious from table 3.

Lines 156-157 ‘Regardless of the specific cause(s) of their non-adherence (16-22), younger patients (in particular adolescents………’

Elaborate on this statement. List plausible reasons for adolescent non-adherence to pharmaco-therapy.

Line 196, typographical error ‘country’ not Country

Line 200, ‘only’ has been repeated twice.

Author Response

Thankyou very much to Reviewer #2 for defining the manuscript "well-written and  contributing to new evidences. Thankyou also for inviting to commend Autjors and  for inviting our group to contribute with further studies.

Comments:

Abstract

1) line 14: "ICS  ....": ok, modified

2) line 16: ok , "spirometric"  added

3) line 20: ok modified according to your suggestion

4) "ds" has been modified in "DS" all over the manuscript

5) line 29: OK, the typographical error has been modified 

Main Text

1) line 56: OK,  "12-month extimation" modified

2 line 87 "Text size .."  modified

3) line 106: "fullfilling": modified

4) Results: the name of statistical test used has been added to all tales and to the figure

5) line 134  - Table 2: SD modified

6) Table 2: you were right: the same data were reported 3 times: probably a mistake when the manuscript was composed for proofs. OK, deleted

7) the outcome was not assessed at three points: the procedure is described in lines 80-+82

8) Line 145: Delete the fig.1: I agree with you that the numbers are reported in Results, but it was decided to maintain the fig.1 for facilitating and stressing the immediate message emerging from the study: such as the dramatic economic impact of non-adherence. In other words, ourdecision was strategic.

9) lines 156-157:  "regardless....": the main causes were added: the corresponding literature was alredy cited

10) line 196: OK, typographical error modified

Finally, the text was revised extensively and several typo errors werfound and modified accordingly

All change are reported in RED

Round 2

Reviewer 1 Report

Manuscript: The extra-cost due to the non-adherence to inhaled treatments in adolescents with mild-to-moderate persistent asthma

The Authors made a partial correction of the manuscript.

Still, the discussion is not satisfactory. Please write in line 159 what is the main finding of the study.

Please standardise the vocabulary of "not-adherence" and "non-adherence".

Please add descriptions of the abbreviations used in the tables below the tables.

Author Response

Dear Reviewer,

we followed point-by-point the 2nd round of your advices and suggestions:

1) the text has been further improved and D has been modified and implemented from the beginning;

2) the terms "non-adherence" (such as a noun) and "not-adherent" (such as an adjective) have been checked and standardized in the text: you can find their righ writing marked in blue and in yellow, repectively

3) even though described in Methods extensively, descriptions of abbrevations have been also reported below  table  1 and Figure 1

R W Dal Negro
